# Controlled Growth of WO_3_ Photoanode under Various pH Conditions for Efficient Photoelectrochemical Performance

**DOI:** 10.3390/nano14010008

**Published:** 2023-12-19

**Authors:** Seung-Je Yoo, Dohyun Kim, Seong-Ho Baek

**Affiliations:** Department of Energy Engineering, Dankook University, Cheonan 31116, Republic of Korea; ysj3802@gmail.com (S.-J.Y.); kimoh422@gmail.com (D.K.)

**Keywords:** tungsten trioxide, tungsten trioxide hydrate, photoanode, water splitting, green hydrogen

## Abstract

Herein, the effects of the precursor solution’s acidity level on the crystal structure, morphology, nucleation, and growth of WO_3_·nH_2_O and WO_3_ nanostructures are reported. Structural investigations on WO_3_·nH_2_O using X-ray diffraction and Fourier–transform infrared spectroscopy confirm that the quantity of hydrate groups increases due to the interaction between H^+^ and water molecules with increasing HCl volume. Surface analysis results support our claim that the evolution of grain size, surface roughness, and growth direction on WO_3_·nH_2_O and WO_3_ nanostructures rely on the precursor solution’s pH value. Consequently, the photocurrent density of a WO_3_ photoanode using a HCl-5 mL sample achieves the best results with 0.9 mA/cm^2^ at 1.23 V vs. a reversible hydrogen electrode (RHE). We suggest that the improved photocurrent density of the HCl-5 mL sample is due to the efficient light absorption from the densely grown WO_3_ nanoplates on a substrate and that its excellent charge transport kinetics originate from the large surface area, low charge transfer resistance, and fast ion diffusion through the photoanode/electrolyte interface.

## 1. Introduction

As the climate crisis has been aggravated because of the excessive use of fossil fuels, the demand for green and renewable energy has steadily increased worldwide [1]. As a result, it is crucial to secure carbon-free energy resources to meet the growing energy consumption. Green hydrogen production is an efficient way of generating high energy density, clean, and storable fuel [2,3]. One of the promising methods of achieving green hydrogen is to adopt a photoelectrochemical route that operates without an external bias [4,5,6,7].

Tungsten trioxide hydrates (WO_3_·nH_2_O, n = 0.33, 1, or 2) and tungsten trioxides (WO_3_) have been widely investigated due to their diverse crystal structures and intriguing electrochemical and photovoltaic properties [8,9,10,11,12,13,14,15,16,17,18,19,20]. In particular, WO_3_ has been considered as a promising semiconductor material for water splitting applications via a photoelectrochemical method, because it is stable and non-toxic, and its band gap is also suitable for visible light absorption [12,14,16,21,22,23]. In general, WO_3_·nH_2_O has been prepared through the solution-phase synthesis, and the WO_3_ has been obtained after removing water molecules from WO_3_·nH_2_O using calcination processes [24,25,26]. Thus, many efforts have been exerted on WO_3_·nH_2_O and WO_3_ nanostructures to develop suitable preparation methods for efficient photoelectrode applications, including acid precipitation and a hydrothermal process, by controlling the morphology, crystal structure, and growth direction at room or moderate temperatures [14,15,16,17,18,19,20,21,27,28]. However, to the best of our knowledge, there has been no report on the relationship between the structural evolution of WO_3_·nH_2_O and its effect on the electrochemical performance of WO_3_ photoanodes while controlling the crystal structure of WO_3_·nH_2_O.

In the present study, we prepared WO_3_·nH_2_O nanostructures by acid precipitation with various levels of hydrochloric (HCl) acid concentration and elucidated the influence of the precursor solution’s acidity level on the crystal structure, morphology, nucleation, and initial growth mode of the WO_3_·nH_2_O samples. Furthermore, we investigated the relationship between the structural evolution of WO_3_·nH_2_O nanostructures and their effect on the water oxidation properties of WO_3_ photoanodes.

## 2. Experiments

### 2.1. Synthesis of WO_3_·nH_2_O and WO_3_ Materials

In order to improve the statistical significance of the results, three samples were prepared by repeating the following experimental procedure three times for each sample. Firstly, WO_3_·nH_2_O was prepared by simple acid precipitation at room temperature with the following processes: 0.4 mmol of sodium tungstate dihydrate (Na_2_WO_4_∙2H_2_O) was dissolved in 15 mL of deionized (DI) water and various amounts of 3 M HCl were dropped in the prepared solution while it was stirred. The samples obtained after adding each volume of HCl solution were labeled HCl-1 mL, HCl-2.5 mL, HCl-5 mL, HCl-7.5 mL, and HCl-10 mL, respectively. Then, 0.8 mmol of ammonium oxalate was dissolved in 15 mL of DI and added to the above solution with stirring to achieve the polymeric oxide. The prepared solution was poured into a Teflon-lined autoclave for hydrothermal synthesis at 140 °C for 3 h. Lastly, the as-grown WO_3_·nH_2_O samples were rinsed with DI and ethyl alcohol several times, and WO_3_ materials were obtained after they were calcined using a furnace at 500 °C in air for 2 h. The preparation procedures of WO_3_·nH_2_O and WO_3_ materials are presented as a schematic illustration in the Appendix A.

### 2.2. Preparation of a WO3 Photoanode on Substrate

Fluorine-doped tin oxide (FTO) glass was used as a substrate cut to 2 cm x 2 cm and then cleaned by sequential sonication in ethyl alcohol, isopropyl alcohol, and DI water for 10 min in each solution. The cleaned FTO glass was vertically aligned on the jig and placed in the autoclave for the growth of the WO_3_ film on the substrate. The detailed preparation procedures of the WO_3_ photoanode on the FTO substrate are provided as an illustration in Appendix A.

### 2.3. Characterizations

The crystal structure was analyzed by X-ray diffraction (XRD, Miniflex 600) using a Cu Kα source. Fourier–transform infrared (FTIR, Perkin Elmer, Waltham, MA, USA) spectroscopy was adopted to study the chemical compositions. Surface analyses were conducted using scanning electron microscopy (SEM, Gemini II, Oberkochen, Germany) and atomic force microscopy (AFM, XE-100). The pH value of the solution was measured with a pH meter (HI2214, HANNA Instruments, Smithfield, RI, USA) which was calibrated with three kinds of pH buffer solutions for use in the high acidity condition. The pH controller was sequentially calibrated in pH 7, pH 4, and pH 1.68 buffer solutions (Thermo Scientific, Waltham, MA, USA). The acidity of the precursor suspension was varied within a wide pH range from 0.05 ± 0.01. to 1.12 ± 0.02 by adjusting the HCl volume. The names of the samples according to the HCl volume and average pH values for precursor suspensions are provided in Table 1. All electrochemical performances were measured with a three-electrode system using platinum as a counter electrode, Ag/AgCl as a reference electrode, and 0.5 M Na_2_SO_4_ (pH 6.9) used as an electrolyte. A Xenon lamp (LS150, ABET technologies) with a power of 150 W was used as the light source, and the illumination area of the photoanode was 2.54 cm^2^. The light intensity of a solar simulator with an air mass (AM) 1.5 G filter was calibrated to a 1 sun condition (100 mW/cm^2^) using a Si photodiode (RR_227 KG5, ABET technologies). And, the photoelectrochemical measurements on the WO_3_ photoanode were conducted by using back-side illumination. An electrochemical workstation (Versastat3, Ametek) was used for linear scan voltammetry (LSV) measurements with a scan rate of 10 mV/s. Electrochemical impedance spectroscopy (EIS) was conducted in the frequency range from 1 MHz to 0.1 Hz with a 10 mV voltage amplitude. The obtained potential of the working electrode was converted into the potential of the reversible hydrogen electrode (RHE) using the following equation: E_RHE_ = E_Ag/AgCl_ + 0.197 + 0.059 pH
where E_RHE_ is the potential referred to for the RHE and E_Ag/AgCl_ is the achieved potential against the Ag/AgCl (3 M KCl saturated) reference electrode.

### 2.4. Data Acquisition

To demonstrate the reproducibility of all the samples, XRD, LSV, and EIS data were obtained. Three samples with different HCl volumes were measured under the same condition and showed almost identical results as shown in the Appendix A; therefore, samples denoted as #1 among the replica were reported in the Results and Discussion section to provide their typical results.

## 3. Results and Discussion

Figure 1 illustrates the structural evolution of WO_3_·nH_2_O by varying the HCl concentration during the acid precipitation process using Na_2_WO_4_ as a precursor, DI water as a solvent, and HCl solution as a precipitant. When HCl was introduced to Na_2_WO_4_, NaCl and tungstic acid (H_2_WO_4_) were generated by the following chemical reaction (1) [18]:2HCl + Na_2_WO_4_ → H_2_WO_4_ + 2NaCl(1)

In general, H_2_WO_4_ can exist in the form of neutral WO_2_(OH)_2_. Under highly acidic conditions, namely, a high H^+^ ion concentration and low pH value, nucleophilic water molecules tend to interact with [WO_2_(OH)_2_] by the following reaction (2) [18,29,30]:[WO_2_(OH)_2_] + 2H_2_O → [WO(OH)_4_(OH_2_)](2)

According to the above reaction, WO(OH)_4_(OH_2_) metal complexes were developed, and the aggregation between adjacent octahedral WO(OH)_4_(OH_2_) occurred in the form of corner-sharing. Moreover, the 2D stacked layers of WO_3_·H_2_O and WO_3_·2H_2_O were gradually created with increasing interaction between seed WO(OH)_4_(OH_2_) and water molecules owing to van der Waals forces [15,31]. 

According to the chemical reactions of (1) and (2), the structural change of the WO_3_·nH_2_O caused by the H^+^ concentration was clearly observed in the XRD patterns as shown in Figure 2. The XRD peaks of each sample corresponded to the hexagonal WO_3_∙0.33H_2_O (JCPDS no. 35-1001), orthorhombic WO_3_∙H_2_O (JCPDS no. 43-0679), and monoclinic WO_3_∙2H_2_O phases (JCPDS no. 18-1420). For the HCl-1 mL sample, 1 mL of HCl was added into a Na_2_WO_4_ solution during acid precipitation, and the hexagonal WO_3_∙0.33H_2_O and orthorhombic WO_3_∙H_2_O phases developed because of the weak interaction between a low H^+^ concentration and nucleophilic water molecules. However, the peak intensity at 22.9°, 24.3°, 26.9°, and 28.3° assigned to the lattice planes of (001), (110), (101), and (200) of the hexagonal structure greatly decreased with an increasing HCl volume, suggesting that the WO_3_∙0.33H_2_O hexagonal phase transformed into the WO_3_∙H_2_O orthorhombic phase as shown in Figure 2b [32]. Likewise, Figure 2b clearly shows that the crystal structure of the HCl-7.5 mL and HCl-10 mL samples were transformed from the orthorhombic WO_3_∙H_2_O phase to the monoclinic WO_3_∙2H_2_O phase owing to the promoted interaction of H^+^ and water molecules leading to the formation of WO_3_∙2H_2_O. Furthermore, based on XRD measurements, we claim that the reproducibility of WO_3_·nH_2_O can be demonstrated as shown in Appendix A. Therefore, we suggest that the crystal structure of WO_3_·nH_2_O (n = 0.33, 1.00, or 2.00) can be precisely engineered by changing the volume of HCl precipitants during room temperature wet synthesis. 

Figure 3 shows the FTIR spectra of the prepared WO_3_·nH_2_O samples under different HCl concentrations. The absorption bands in the range of 500–1000 cm^−1^ are characteristic of the various WO_3_ crystal lattices, i.e., W–O, O–W–O, and W=O bonds. The strong peak around 630 cm^−1^ was assigned as the stretching mode of metal–oxygen (O–W–O) [15,33]. In addition, the strong IR absorption observed at 945 cm^−1^ was due to the stretching mode of W=O bonds of hydrated WO_3_ with an increasing H^+^ concentration [33]. Two strong absorption peaks near 1610 and 3500 cm^−1^ arose from the H–O–H bending and O–H stretching vibrations of H_2_O, respectively, suggesting that WO_3_·nH_2_O can host a large number of water molecules either in the coordinated or interlayer form [33,34]. Thus, the FTIR results suggested that the quantity of hydrate groups increased due to the interaction of H^+^ and nucleophilic water molecules with an increasing HCl volume, which is consistent with the XRD results.

To elucidate the relationship between the crystal structure and the initial nucleation growth of the WO_3_·nH_2_O on an FTO substrate, AFM was used to achieve three-dimensional (3D) images of all the samples. The 3D AFM images of the WO_3_·nH_2_O nanostructures grown on FTO glass with different amounts of HCl during acid precipitation are presented in Figure 4. The AFM image of the bare FTO glass exhibits a smooth surface, while island-like grains can be observed for the samples with different HCl concentrations. The smooth surface of bare FTO glass exhibits a root mean square (RMS) of 38.4 nm (Figure 4a). However, as the amount of HCl increased during acid precipitation of WO_3_·nH_2_O nanostructures, sharp grains developed on the surfaces of the FTO glass (Figure 4b–f). The size and RMS of these grains increased up to a HCl volume of 5 mL as shown in Figure 4b–d. Among the prepared samples, the HCl-5 mL sample shows higher values of RMS surface roughness, indicating that the growth rate of WO_3_·nH_2_O nanostructures is the highest among the samples. In contrast, the growth rate of WO_3_·nH_2_O was decreased in the samples prepared with 7.5 mL and 10 mL of HCl. The driving force for nucleation is closely related to the supersaturation of WO(OH)_4_(OH_2_) seed molecules according to the chemical reaction of (2). At a larger pH value (low H^+^ concentration), the number of WO_3_ nuclei could be limited because of the small supersaturation of seed molecules. On the contrary, at a smaller pH value (high H^+^ concentration), a great deal of nuclei could be quickly generated, due to the larger supersaturation of seed molecules. However, Pham et al. report that the condensed H^+^ ion had a confining effect that generates a strong electric field, inhibiting long-ordered nucleation growth [18]. Therefore, the optimized nucleation growth of WO_3_ nanoplates can be obtained in the moderate pH value and it is found to be the HCl-5 mL sample in our experimental results. Therefore, we suggest that the evolution of grain size, surface roughness, and inter-grain space distribution on WO_3_·nH_2_O nanostructures rely on the precursor solution’s acidity level.

After the hydrothermal growth and calcination process, WO_3_·nH_2_O nanostructures can be dehydrated and transformed to achieve the WO_3_ phase [26,27]. To study the crystal structure of WO_3_ materials on FTO glass, we conducted the XRD measurements, and the diffraction results are provided in Figure 5a. Regardless of the WO_3_·nH_2_O nanostructures and HCl concentration, the crystal structure of all the samples was identified as monoclinic WO_3_ (JCPDS no.43-1035) after calcination [35]. It can be seen from Appendix A that these results are demonstrated via repeated XRD measurements using a dummy sample. It should be noted that the peak intensity of the crystal plane gradually increased from the (002) to the (200) plane, as the HCl volume increased from 1 to 5 mL. The highest diffraction peak intensity of the crystal plane (200) was obtained from the WO_3_ nanoplate on FTO glass with the incorporation of 5 mL of HCl as shown in Figure 5b. 

Figure 6 shows the SEM images of the monoclinic WO_3_ nanoplates grown on the FTO glass substrate after hydrothermal growth and the calcination process. The low-magnification top-view SEM images (Figure 6a–e) reveal that the HCl volume significantly affects the density of the WO_3_ nanoplate. As the HCl was incorporated into the precursor solution, the density of the WO_3_ nanoplates increased up to the HCl-5 mL sample; afterward, the density of the WO_3_ nanoplates drastically decreased in the WO_3_ nanoplates prepared with 7.5 mL and 10 mL of HCl, according to the AFM results as provided in Figure 4. Even though the WO_3_ nanoplate samples prepared by 1 mL of HCl had the lowest density among all the samples, due to the low supersaturation of seed molecules, a high-magnification SEM image (Figure 6f) revealed that the largest WO_3_ nanoplates were created by subsequent oxolation reactions on WO_3_ nuclei. Meanwhile, as the HCl volume increases, the seed molecules become more supersaturated, which makes it possible to generate a lot of nuclei on the substrate. However, the condensed H^+^ ion generates a strong electric field, inhibiting long-ordered nucleation and subsequent WO_3_ nanoplate growth as shown in Figure 6j. These results support our claim that the crystal growth of the WO_3_ nanoplate was greatly influenced by the initial nuclei density of WO_3_·nH_2_O nanostructures on the substrate; therefore, the optimum pH value for the growth of WO_3_ nanoplates should lie in the moderate pH range and it is revealed to be the HCl-5 mL sample as shown in Figure 6h. Figure 6k–o show the cross-sectional SEM images of the WO_3_ nanoplates on the substrates. Vertically aligned WO_3_ nanostructures with a length of 1–2 μm were well-grown on the FTO layer as indicated in Figure 6k. As confirmed from the magnified top-view SEM images, the largest WO_3_ nanoplates were observed in the HCl-1 mL sample (Figure 6k). On the other hand, it should be noted that the rest four samples have a similar growth thickness, but they have a different density of WO_3_ nanoplates. Moreover, we clearly observed that the growth direction of the WO_3_ nanoplates changed from a slanted to a vertical direction for the substrates according to the proportion of the WO_3_ nanoplates (Figure 6k–o). When the WO_3_ nanoplates were sparsely distributed on the substrate under a low acidity level, they were grown on FTO glass with an inclined direction against the substrate. However, as the number of WO_3_ nanoplates increased at high acidity levels, they densely grew on the substrate and the density of WO_3_ nanoplates was maximized in the HCl-5 mL sample as shown in Figure 6m. 

Consequently, the preferential growth direction of the WO_3_ nanoplate changed according to the initial growth conditions, which resulted in a change in the strongest XRD peak from (002) to (200) as shown in Figure 5b. Thus, we claim that the HCl volume plays a crucial role in the growth process of WO_3_·nH_2_O nanostructures and WO_3_ nanoplates on FTO substrates. 

The PEC performance of the prepared WO_3_ photoanodes was investigated under AM 1.5 G conditions with an illuminated area of 2.54 cm^2^. Figure 7 shows the LSV results of the WO_3_ photoanodes with different amounts of HCl in a 0.5 M Na_2_SO_4_ electrolyte. According to the LSV curve, the dark currents of all the samples were less than 20 μA/cm^2^ and were regarded as negligible. When the HCl concentration increased, the photocurrent density was enhanced as shown in Figure 7. The photocurrent density of the photoanode reached the highest value with 0.9 mA/cm^2^ at 1.23 V vs. RHE for the HCl-5 mL sample, while that of the HCl-1 mL sample exhibited the lowest with 0.3 mA/cm^2^ at the same voltage. However, as the incorporated amount of HCl further increased to more than 5 mL, the photocurrent densities decreased from 0.65 to 0.57 mA/cm^2^ at 1.23 V vs. RHE for the WO_3_ photoanodes prepared with 7.5 mL and 10 mL of HCl. As a result of repeated measurement using a replica, we suggest that the WO_3_ photoanodes with different HCl volumes can be identically reproduced as shown in Appendix A. Therefore, we claim that the improved photocurrent is due to the enhanced light absorption from the dense and large volume of WO_3_ nanoplates in a HCl-5 mL photoanode, which facilitates the charge transport kinetics by producing more photogenerated electron-hole pairs. 

EIS measurements were conducted to prove the charge transfer kinetics at the WO_3_ photoanodes–electrolyte interface. The EIS results are presented as a Nyquist plot at an open-circuit voltage under AM 1.5 G illumination as shown in Figure 8. As seen in Figure 8, the HCl-5 mL photoanode sample represents the smallest semicircle compared to the other electrodes, indicating a much lower photoelectrode–electrolyte interfacial resistance compared to the other electrodes [36]. In this work, the obtained EIS data were fitted into the RC circuit model as provided in the inset of Figure 8, containing a resistance and a capacitance in parallel. Because the water oxidation takes place at the WO_3_ photoanode–electrolyte interface, the charge transfer resistance (R_ct_) and double layer capacitance (CPE) can be observed at the interface [37,38]. The charge transfer resistances for all the samples were obtained by using the ZSimpWin software (version 3.21), and their values for the HCl-1 mL, HCl-2.5 mL, HCl-5 mL, HCl-7.5 mL, and HCl-10 mL samples were 1445, 1372, 687, 987, and 2704 Ω, respectively. The fitting results clearly confirm that the derived R_ct_ values for the HCl-5 mL photoanode were the smallest among all the samples. From the results of EIS data for three samples under the same condition, we confirm that the reproducibility of photoanodes–electrolyte interfacial resistance can be demonstrated as shown in Appendix A. Therefore, we suggest that the superior PEC performance of the HCl-5 mL sample is attributed to the electrode–electrolyte accessibility of the WO_3_ nanoplate owing to the densely grown nanostructure, efficient charge transfer, and fast ion diffusion between the photoanode and electrolyte interface.

## 4. Conclusions

In summary, we demonstrated the influence of the precursor solution’s pH level on the structural evolution, crystal growth, and nucleation of WO_3_·nH_2_O and WO_3_ nanostructures. The XRD and FTIR results on WO_3_·nH_2_O verified that the quantity of hydrate groups increased due to the interaction of H^+^ and water molecules with an increasing HCl volume. Surface analyses using AFM and SEM confirmed that the evolution of grain size, surface roughness, and grain growth on WO_3_·nH_2_O and WO_3_ nanostructures were affected by the precursor solution’s pH value. The photoelectrochemical performance confirmed that the best photocurrent density and charge transfer resistance were obtained using the HCl-5 mL sample due to its structural benefits, which facilitated efficient charge transfer and ion diffusion between the photoanode and the electrolyte. Our findings offer a facile approach to obtaining superior performance in PEC water splitting. 

## Figures and Tables

**Figure 1 nanomaterials-14-00008-f001:**
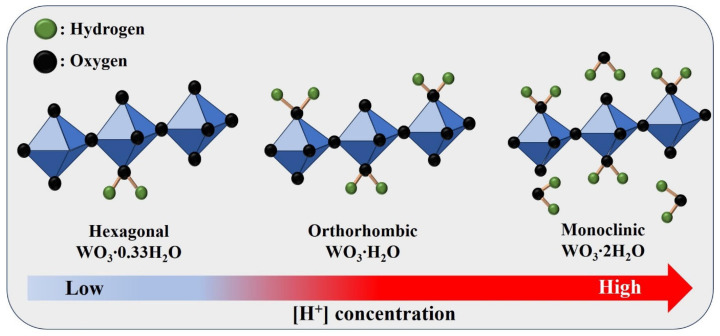
Schematic diagram of the WO_3_·nH_2_O under various pH conditions during the acid precipitation process.

**Figure 2 nanomaterials-14-00008-f002:**
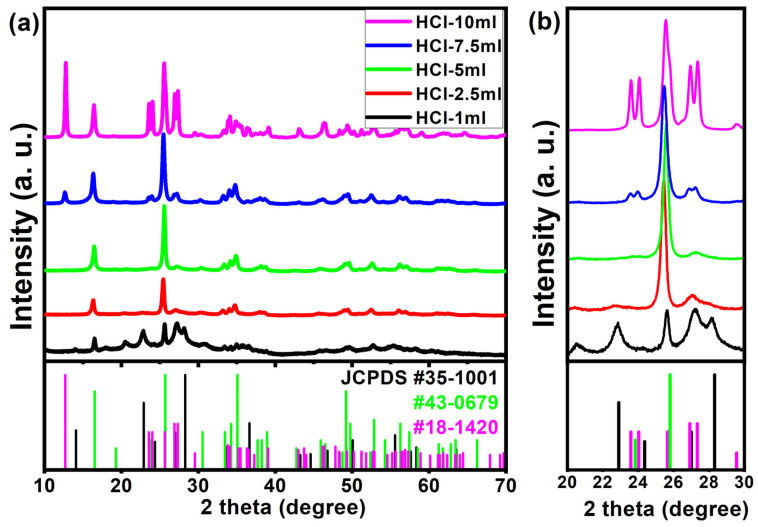
XRD spectra of the WO_3_·nH_2_O powders obtained by an acid precipitation process. (**a**) 2theta range from 10° to 70° and (**b**) magnified XRD spectra in the 2theta range from 20° to 30°.

**Figure 3 nanomaterials-14-00008-f003:**
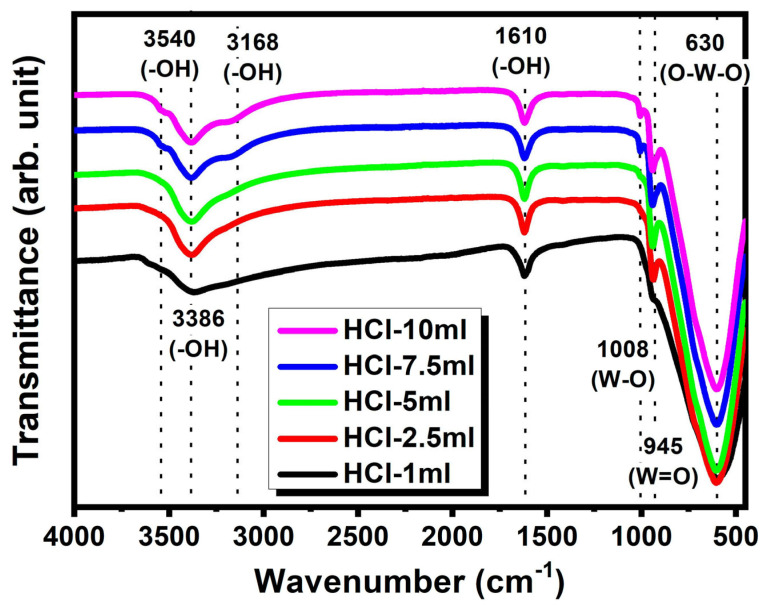
FTIR spectra of the WO_3_·nH_2_O powders obtained by an acid precipitation process.

**Figure 4 nanomaterials-14-00008-f004:**
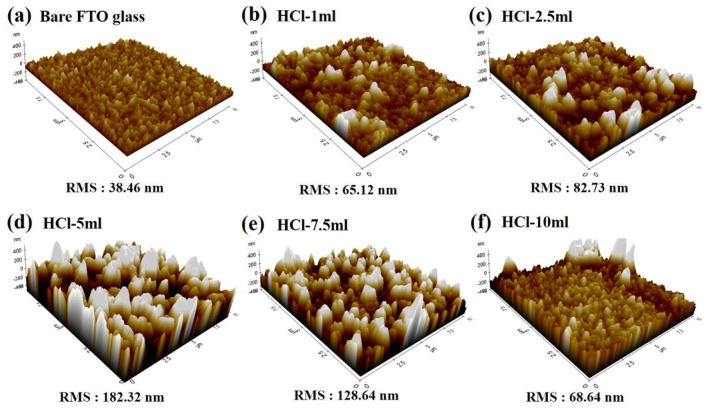
AFM images of the WO_3_·nH_2_O powders grown on FTO substrates with different HCl volumes during acid precipitation.

**Figure 5 nanomaterials-14-00008-f005:**
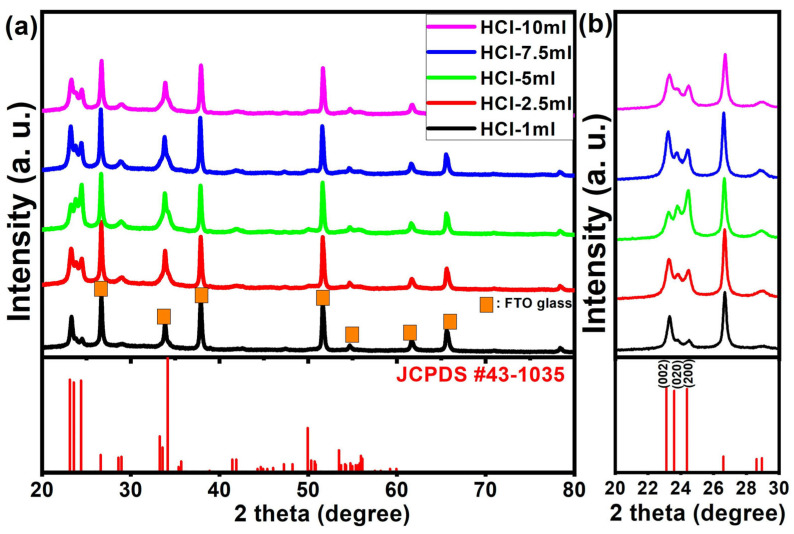
XRD results of the WO_3_ nanoplates grown on FTO glass. (**a**) 2theta range from 20° to 80° and (**b**) magnified XRD spectra in the 2theta range from 20° to 30° (orange squares indicate the XRD peak positions of the FTO substrate).

**Figure 6 nanomaterials-14-00008-f006:**
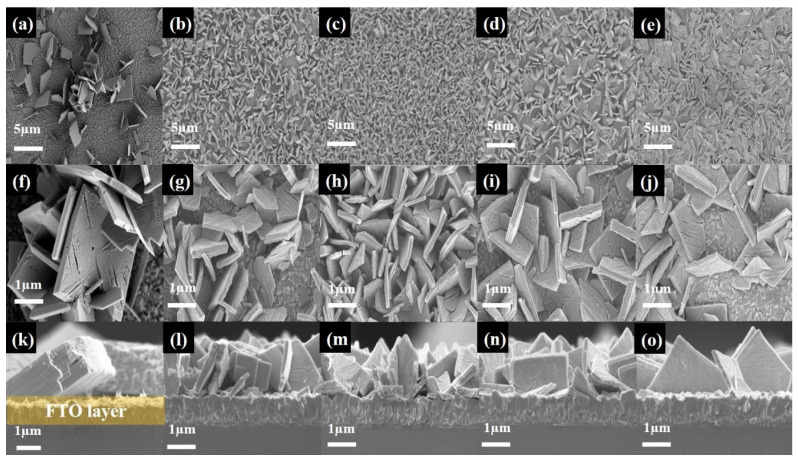
SEM images of the WO_3_ nanoplates grown on FTO glass with different HCl volumes. (**a**) HCl-1 mL, (**b**) HCl-2.5 mL, (**c**) HCl-5 mL, (**d**) HCl-7.5 mL, (**e**) HCl-10 mL samples. Magnified (**f**–**j**) top-view and (**k**–**o**) cross-section view SEM images of the WO_3_ nanoplates corresponding to the upper panel.

**Figure 7 nanomaterials-14-00008-f007:**
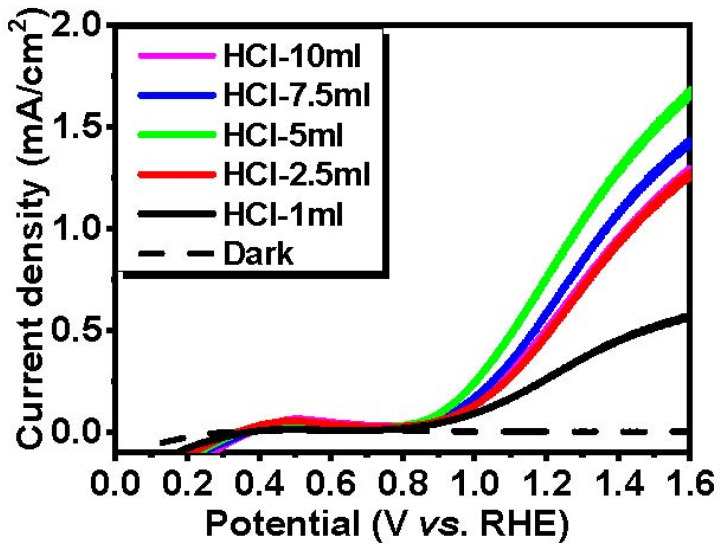
LSV curves obtained from the WO_3_ photoanodes under AM 1.5 G conditions.

**Figure 8 nanomaterials-14-00008-f008:**
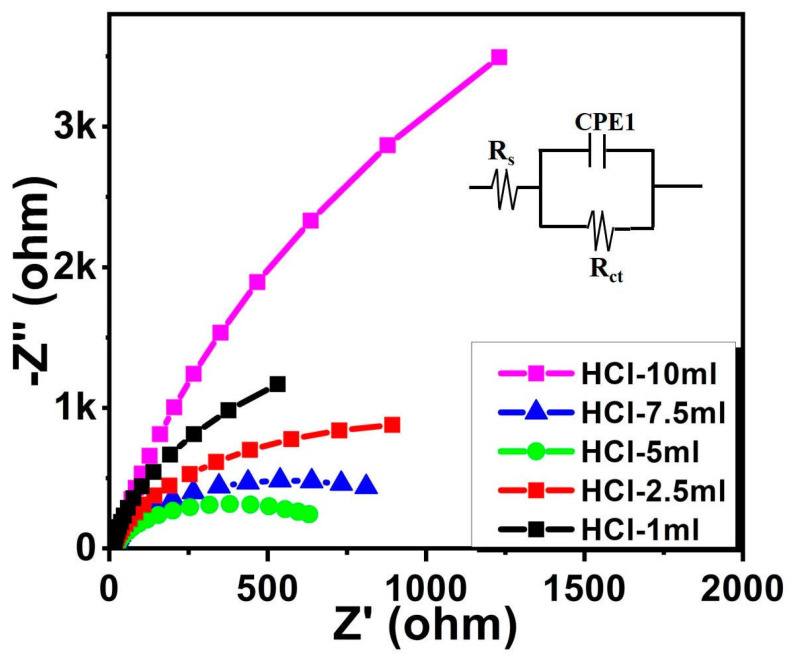
EIS Nyquist plots of all the photoanodes under AM 1.5 G conditions. (Inset: the equivalent circuit model).

**Table 1 nanomaterials-14-00008-t001:** Average pH values of precursor solutions according to HCl volume.

Sample	HCl-1 mL	HCl-2.5 mL	HCl-5 mL	HCl-7.5 mL	HCl-10 mL
pH value	1.12 ± 0.02	0.86 ± 0.02	0.62 ±0.01	0.34 ± 0.01	0.05 ± 0.01

## Data Availability

The data presented in this study are available on reasonable request from the corresponding author.

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
