# Peer review of "Controlled Growth of WO3 Photoanode under Various pH Conditions for Efficient Photoelectrochemical Performance"

_nanomaterials, 2023, doi:10.3390/nano14010008_

Round 1
Reviewer 1 Report
Comments and Suggestions for Authors
In this work, the authors demonstrated the influence of the precursor solution’s pH level on the structural evolution, crystal growth, and nucleation of WO3⋅nH2O and WO3 nanostructures. The XRD and FTIR results on WO3⋅nH2O verified that the hydrate groups increased due to the interaction of H+ and water molecules with an increasing HCl volume. Surface analyses through AFM and SEM confirmed that the evolution of grain size, surface roughness and grain growth on WO3∙nH2O and WO3 nanostructures were affected by the precursor solution’s pH value. Overall, the manuscript is nicely written, and the data is well-presented. Please see below a few points that need to be addressed.
Comments:
1. The photoelectrochemical performance confirmed that the best photocurrent density and charge transfer resistance were obtained using the HCl–5ml sample, and the reasons for this good performance should be given.
2. The photocurrent density of a WO3 photoanode using a HCl–5ml sample achieves the best results with 0.9 mA/cm2 at 1.23 V vs, how about their stability under high temperature and high humidity?
3. WO3 has also been reported in perovskite solar cells, such as, 10.1007/s12596-022-01035-3; 10.1002/solr.202300438;10.1039/C8TA08287A.
4. Fig. 2(b) clearly shows that the crystal structure of the HCl–7.5ml and HCl–10ml samples were transformed from the orthorhombic WO3∙H2O phase to the monoclinic WO3∙2H2O phase, why?
Comments on the Quality of English Language
no comments
Author Response
Dear Reviewer,
We appreciate the reviewers’ constructive comments and efforts that helped to considerably improve the manuscript. We have carefully studied and addressed all the comments given by the reviewers. In the following sections, all responses are described in blue font for each reviewer’s comments. The detailed corrections are described in the “Revised manuscript” file in red color.
Please see the attachment.
Thank you.

Reviewer 2 Report
Comments and Suggestions for Authors
A relevant study is presented on the preparation of WO3 electrodes that can be of interest for applications in electrochemical cells. The mechanistic explanations look sound, but details are needed to support the arguments. Most important point, the statistical significance of the results has to be proven. As of now, the results may as well have been produced by coincidence. The following comments should help in such revision:
1. In the experimental section, it should be listed how many replicas of each film type were prepared. Were the best, a typical or the average values reported for each condition?
2. Details are needed in the experimental section: how were the films formed (e.g., 15 ml of solution used for one substrate?), which buffer solutions were used for calibration (should be more than one to cover a range, needs to be adjusted to very high acidity in this case)? Calibration of pH cannot be correct since an activity coefficient as low as 0.36 would be needed in order to explain a pH of 1.14 for a 3/16 M HCl. This seems too low. Oppositely, an activity coefficient larger 1 would be needed to explain pH=0.04 for 30/25 M. Therefore, the pH-values provided seem to be problematic if my rough calculations were correct. Manufacturers of equipment should be listed, measurement parameters provided.
3. A growth mechanism should be discussed in order to explain the assumed pH-dependence. It should be discussed, which influence is seen for the concentration of the W precursor.
4. Not just the roughness, also the film thickness should be discussed.
5. Differences in XRD need to be reproduced in order to claim significant correlation to pH of the precursor solution. Same holds for photoelectrochemical measurements (current-voltage and impedance).
6. The photoelectrochemical reaction (water oxidation?) should be named as such. Why is ion diffusion inside the electrode material relevant for it?
7. An equivalent circuit needs to be shown and its relevance for the present electrode/electrolyte discussed. Detailed mechanistic discussions as desired by the authors probably even then, however, will barely be possible.
Author Response

(The authors gave the same response as above.)

Round 2
Reviewer 2 Report
Comments and Suggestions for Authors
The authors invested a great effort to revise their manuscript in response to my earlier comments. I will answer to these changes in the number sequence of the original comments:
1. Reproducibility of data is now taken care of in a good way. However, it should be explicitly stated if the synthesis was performed three times under identical conditions or if three samples were prepared for a given synthesis, performed only once. Further, it should be stated for which properties an average was calculated and for which properties a typical result was reported (and reproducibility proven in the SI), since both approaches were used in the work, which is perfectly fine.
2. It is still not clear to this reviewer how the films were formed. Were the final powders of WO3 (after calcination) used to prepare a paste and then cover the FTO substrates? Were the FTO substrates put into the autoclave?
Still, some manufacturers of equipment and measurement parameters are missing. No indication, e.g., is provided, how the AM 1.5 conditions were controlled. Further, it is not clear from the text if the samples were illuminated from the WO3- (front-) or the glass/FTO (back-) side.
3. It is very helpful to have the reaction equation and ideas of growth included in the paper. However, the text in lines 185-190 repeats larger parts of the text in lines 150-157. High similarity of sentences should be avoided (also true for the mention of three identical samples at different occurrence in the text).
4. Still, there is no information provided about the thickness of the different films. In a photoanode, the amount of absorber might be decisive for the amount of absorbed light and, hence, also for the photocurrent generated. Only in cases for samples thick compared to the active layer thickness AND for front-side illumination, the film thickness may not decide over the amount of effectively absorbed light. Further, however, a thicker film always carries the potential to increase the series resistance. For both reasons, a discussion of the film thickness (or at least, the total amount of WO3 on the substrates) is essential in the present case to assign a reason for the good performance of the “5ml samples”.
5. OK.
6. OK
7. The equivalent circuit is now provided but its motivation stays weak. The text in lines 222-232 appears quite inhomogeneous. No reference is given to prove that the chosen circuit is applicable in this case. Ion diffusion inside the WO3 photoanode as mentioned in line 227 has no specific role in the argument. The reader has no chance to follow the thought. This reviewer does not disagree with the relevance of the circuit, just it would be nice to the readership to indicate its relevance to them.
For these reasons, publication of the work cannot be recommended before further correction.
Author Response
Dear Reviewer,
We appreciate the reviewers’ constructive comments and efforts that helped to considerably improve the manuscript. We have carefully studied and addressed all the comments given by the reviewer. In the following sections, all responses are described in blue font for each reviewer’s comments. The detailed corrections are described in the “Revised manuscript” file in red color.
Please see the attachment.
Thank you.

Round 3
Reviewer 2 Report
Comments and Suggestions for Authors
The authors have now fixed all the problems that I had mentioned.